# ROADBENCH: BENCHMARKING MLLMS ON FINE-GRAINED SPATIAL UNDERSTANDING AND REASONING IN URBAN ROAD SCENARIOS

## ABSTRACT

Multimodal large language models (MLLMs) have demonstrated powerful capabilities in general spatial understanding and reasoning. However, their fine-grained spatial understanding and reasoning capabilities in complex urban scenarios have not received significant attention in the fields of both research and industry. To fill this gap, we focus primarily on road markings as a typical example of fine-grained spatial elements in urban scenarios, given the essential role of the integrated road traffic network they form within cities. Around road markings and urban traffic systems, we propose **RoadBench**, a systematic benchmark that comprehensively evaluates MLLMs' fine-grained spatial understanding and reasoning capabilities using BEV and FPV image inputs. This benchmark comprises six tasks consisting of 9,121 strictly manually verified test cases. These tasks form an systematic evaluation framework that bridges understanding at local spatial scopes to global reasoning. They not only test MLLMs' capabilities in recognition, joint understanding, and reasoning but also assess their ability to integrate image information with domain knowledge. After evaluating 14 mainstream MLLMs, we confirm that RoadBench is a challenging benchmark for MLLMs while revealing significant shortcomings in existing MLLMs' fine-grained spatial understanding and reasoning capabilities within urban scenarios. In certain tasks, their performance even falls short of simple rule-based or random selection baselines. These findings, along with RoadBench itself, will contribute to the comprehensive advancement of spatial understanding capabilities for MLLMs. The benchmark code, example datasets, and raw evaluation results are available at https://anonymous.4open.science/r/RoadBench-A00E.

## 1 INTRODUCTION

Multimodal large language models (MLLMs) have become a crucial tool for recognizing and understanding the real world due to their powerful combined visual-language comprehension and reasoning capabilities (Achiam et al., 2023; Yin et al., 2024; Bai et al., 2025; Team et al., 2025a). They are progressively replacing specialized models in fields such as satellite image recognition (Zhang et al., 2024; 2025b; Feng et al., 2025b), autonomous driving (Cui et al., 2024; Zhang & Nie, 2024; Tian et al., 2025), embodied intelligence (Driess et al., 2023; Li et al., 2024b), etc. Spatial understanding and reasoning in urban environments, as one of the key real-world application scenarios for MLLMs, has also garnered significant attention recently (Roberts et al., 2024; Feng et al., 2025a;b). Various research efforts have released benchmarks (Zhou et al., 2025; Feng et al., 2025c; Xie et al., 2025) for spatial understanding and reasoning tasks in urban environments, evaluating MLLM performance from multiple perspectives. These benchmarks provide the foundational data and evaluation criteria necessary to advance MLLM research for real-world applications.

However, we observe that existing benchmarks on urban scenarios mainly focus on whole-image-level understanding or isolated object recognition of specific types, such as GeoQA, landmark recognition, and vehicle detection. There appears to be a lack of attention to understanding and reasoning about fine-grained spatial elements within urban scenarios. Recently, cutting-edge research has been exploring how to enhance MLLM's fine-grained understanding and reasoning capabilities in scenarios such as mathematical problems (Zhang et al., 2025a) and remote sensing images (Ou et al.,

2025). Nevertheless, studies on MLLM's such capabilities have yet to be applied to urban scenarios, particularly lacking evaluation benchmarks tailored to real-world urban settings.

In urban scenarios, a representative example of such fine-grained spatial structural elements is the markings painted on city road surfaces shown in Figure 1. These narrow and long lines and arrows, drawn based on specialized traffic knowledge, collectively form an integrated system that effectively divides and organizes urban space, thereby regulating the movement of pedestrians and vehicles. The tasks of understanding and reasoning about road markings pose significant challenges to MLLM's capabilities:

- The ability to recognize fine-grained structures at a global scale, as road markings are typically thin and extend across the entire image.
- The joint understanding and reasoning of multiple fine-grained structures, as road markings require overall consideration for accurate semantic recognition.
- The integration of image information and domain knowledge to generate reasonable responses.

We believe that developing MLLM's fine-grained understanding and reasoning capabilities in urban scenarios will not only advance its applications in transportation fields such as high-definition map auto-generation and end-to-end autonomous driving. It will also enhance MLLM's general visual-spatial reasoning abilities, particularly regarding various artificially designed symbol and marking systems in other fields.

Therefore, we propose **RoadBench**, a systematic benchmark primarily designed to evaluate MLLM's understanding and reasoning capabilities regarding fine-grained spatial structural elements in urban scenarios. RoadBench provides a rich collection of annotated Bird's-Eye View (BEV) and First-Person View (FPV) images, comprising a total of 9,121 test cases across 6 tasks. These tasks include not only lane counting and lane designation recognition, which are directly based on understanding and reasoning about road markings, but also extend to road network correction and road type classification tasks that rely on joint reasoning involving both road markings and other information within the image. The test cases comprising Road-Bench originate from manually selected images across multiple Chinese cities. All labels and ground truth data have been labeled and checked by humans to ensure accuracy, with all potentially privacy-sensitive information completely masked and anonymized. These test cases en-

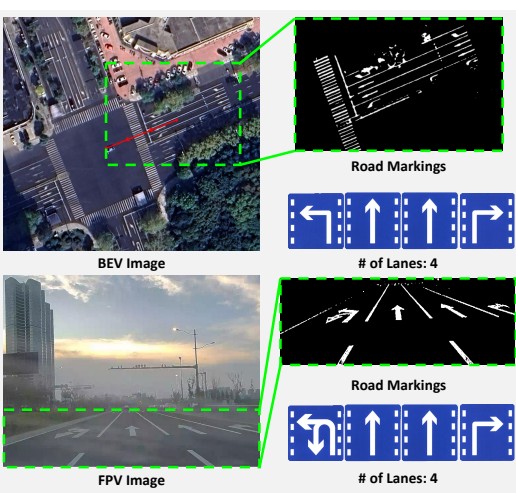

Figure 1: Examples of road markings in BEV and FPV images with their meanings.

compass diverse factors that may impact MLLM understanding, including road and intersection patterns, lighting conditions, seasons, and image resolution. This comprehensively supports the systematic evaluation of MLLM's fine-grained spatial understanding and reasoning capabilities. Based on RoadBench, we conducted a systematic evaluation of 14 mainstream closed-source and open-source MLLMs. Benchmark results indicate that MLLMs failed to achieve satisfactory outcomes across various tasks, even underperforming against baselines based on random choice or simple rules in certain tasks. This highlights both the current limitations of MLLMs in achieving fine-grained spatial understanding and reasoning capabilities within urban environments, and underscores RoadBench as a highly challenging benchmark for MLLMs to evaluate such capabilities.

Overall, the main contributions of this paper include the following points:

- We propose a systematic benchmark named **RoadBench**, designing six tasks under both BEV and FPV image inputs to comprehensively evaluate MLLMs' understanding and reasoning capabilities regarding fine-grained spatial elements in urban scenarios from local to global scopes.
- We collected, processed, and annotated 9,121 test cases as datasets for the six tasks in RoadBench using data sources such as satellite imagery, online map service providers, and crowd-sourced

in-vehicle camera photo databases. All test cases underwent rigorous manual verification and annotation to ensure the accuracy of their labels.

- We conducted a systematic evaluation of 14 mainstream MLLMs using RoadBench. The results demonstrate that RoadBench is a highly challenging benchmark for MLLMs, while also revealing the limitations of existing MLLMs in fine-grained spatial understanding and reasoning capabilities within urban scenarios.

## 2 RELATED WORK

### 2.1 MULTIMODAL LARGE LANGUAGE MODELS FOR SPATIAL INTELLIGENCE

Inspired by the powerful comprehension and complex reasoning capabilities emerging from large language models (Brown et al., 2020; Touvron et al., 2023), researchers have achieved alignment and fusion between text and image modalities through techniques such as CLIP (Radford et al., 2021) and BLIP (Li et al., 2022) to construct multimodal large language models (Achiam et al., 2023; Yin et al., 2024; Bai et al., 2025; Team et al., 2025a). These MLLMs can simultaneously process textual and visual inputs to perform comprehension and reasoning tasks, enabling them to accomplish a wide range of complex operations (Cui et al., 2024; Xiao et al., 2024). Leveraging the capabilities of MLLM foundation models, some researchers have successfully developed MLLMs specifically tailored for urban scenarios, such as CityGPT (Feng et al., 2025a) and UrbanLLaVA (Feng et al., 2025b). Additionally, some recent researchers have focused on enhancing MLLM's fine-grained spatial understanding and reasoning capabilities in scenarios such as mathematical problems (Lu et al., 2023; Zhang et al., 2025a; Wei et al., 2024), daily images (Azzolini et al., 2025; Cheng et al., 2024; Guo et al., 2024; Chen et al., 2024), abstract visual puzzles (Wang et al., 2024; Ramakrishnan et al., 2024), and remote sensing images (Ou et al., 2025). However, research on the fine-grained understanding and reasoning capabilities of MLLM in urban scenarios remains relatively rare at present. The lack of benchmarks may be one of the primary factors constraining such work.

### 2.2 SPATIAL BENCHMARKS FOR MULTIMODAL LARGE LANGUAGE MODELS

In the field of multimodal large language models, researchers have released extensive benchmarks. For example, MME (Fu et al., 2024), Seed-bench (Li et al., 2024a), and MMBench (Liu et al., 2024) introduce various perception, cognition, and reasoning tasks to comprehensively evaluate the performance of MLLMs. Some recent researchers have also developed benchmarks using abstract visual puzzles to evaluate the spatial reasoning capabilities of MLLMs, for example, SPACE (Ramakrishnan et al., 2024), SpatialEval (Wang et al., 2024), and Spatial457 (Wang et al., 2025). In real-world urban scenarios, CityBench (Feng et al., 2025c) primarily evaluates MLLM's ability to perceive and understand images about cities, as well as its planning and decision-making capabilities. UrBench (Zhou et al., 2025) focuses on evaluating MLLMs in cross-view urban scenarios. CityEQA (Zhao et al., 2025) investigates the urban scenarios from the aerial vehicle in a realistic 3D urban simulator. DriveBench (Xie et al., 2025) focuses on evaluating the reliability of MLLM in autonomous driving applications. Overall, there is currently no benchmark attentive to the recognition, understanding, or reasoning of fine-grained spatial elements like road markings in urban scenarios, which limits the evaluation of MLLM capabilities in real world scenarios.

## 3 ROADBENCH

### 3.1 BENCHMARK OVERVIEW

RoadBench is designed as a benchmark to evaluate the fine-grained spatial understanding and reasoning capabilities of MLLM models in urban scenarios. As shown in Figure 2, RoadBench contains two types of urban scene images: bird's eye view images derived from satellite imagery and first-person view images captured by in-vehicle cameras. Based on these two categories of images, we design six benchmark tasks centered around road markings, which are common fine-grained spatial elements in urban scenarios. These tasks are organized in order of the spatial scope from local to global that MLLM requires for task completion, collectively forming a hierarchical and systematic benchmark that spans from fine-grained local perception and understanding to global contextual

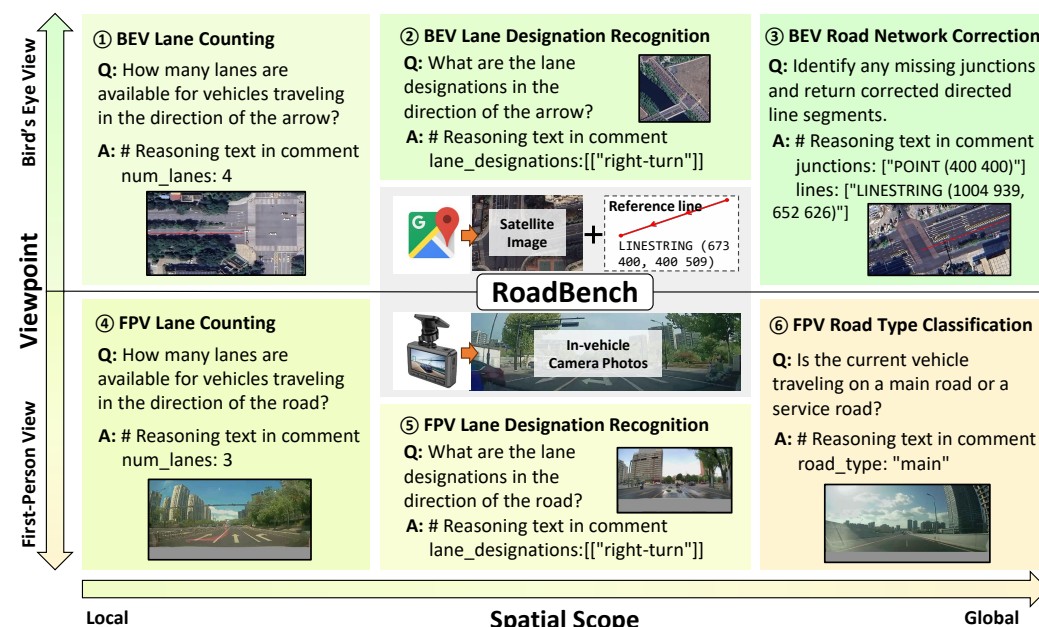

Figure 2: The overview of RoadBench.

Table 1: Dataset statistics and evaluation metrics for the proposed RoadBench.

| Task Category | View | Evaluation Metrics | # Test Cases |
|---|---|---|---|
| BEV Lane Counting | BEV | Precision, Recall, F1-Score, RMSE | 2,908 |
| BEV Lane Designation Recognition | BEV | Hamming Loss, Acc. | 2,908 |
| BEV Road Network Correction | BEV | RMSE, Fréchet Distance | 840 |
| FPV Lane Counting | FPV | Precision, Recall, F1-Score, RMSE | 737 |
| FPV Lane Designation Recognition | FPV | Hamming Loss, Acc. | 737 |
| FPV Road Type Classification | FPV | Acc. | 991 |
| **Total** | | | **9,121** |

reasoning. The number of test cases included in each task is listed in Table 1. The entire benchmark comprises a total of 9,121 test cases. These rigorously hand-curated and processed test cases cover diverse scenarios involving different road and intersection patterns, lighting conditions, seasons, and image resolutions, providing comprehensive coverage of the diversity in urban environments.

## 3.2 BENCHMARK TASKS

The six benchmark tasks in RoadBench include lane counting and lane designation recognition using BEV images and FPV images respectively, as well as road network correction using BEV images and road type classification using FPV images. Each task's setup and evaluation method will be introduced below. The specific MLLM prompts used, typical MLLM input images, and MLLM responses can be found in Appendix D.

**BEV Lane Counting.** The lane counting task based on BEV images requires the MLLM to determine the number of lanes contained within the road indicated by the reference line, using the input satellite image and the additional directed reference line. As shown in the example in Appendix D.1, reference lines are drawn as polylines with prominent red arrows on satellite imagery to serve as visual prompts. The MLLM's output is required to be a text describing the reasoning process along with the number of lanes. This task aims to conduct a preliminary evaluation of MLLM's ability to follow both visual and textual prompts simultaneously while analyzing and understanding the narrow road markings surrounding reference lines in satellite imagery. The coordinates of reference lines and actual lane counts in test cases originate from the databases of a tier-1 online map service provider, which also supplies this data for online service delivery. In terms of evaluation metrics, we employed multi-class

classification metrics (Precision, Recall, F1-score) on one hand, while also utilizing RMSE to assess the deviation between MLLM outputs and ground truth.

**BEV Lane Designation Recognition.** The lane designation recognition task based on BEV images shares the same MLLM input setup as the BEV lane counting task, including satellite images and reference lines. In difference, this task further requires MLLM to infer the direction type of each lane by identifying and understanding road surface arrows, among other methods, when the correct number of lanes is known. Lane direction types include U-turn, left turn, straight ahead, right turn, and their combinations. This task not only challenges the MLLM's ability to jointly understand a complete set of road markings, but also tests its domain expertise in lane designation. Ground truth data also comes from the online service database. Given the combinatorial nature of lane directions, we treat the classification of each lane as an independent multi-label classification problem. We employ the Hamming Loss as the primary metric and utilize accuracy as a more stringent metric to evaluate whether MLLMs fully comprehend the lane directions.

**BEV Road Network Correction.** The road network correction task based on BEV images is both highly challenging and of significant practical value. This task begins by feeding an inaccurate reference line into MLLMs. These reference lines often contain errors such as missing actual junctions, which can lead to map services providing incorrect directions to users. MLLMs are tasked with understanding the concepts of junctions and road segments, then inferring the correct junction locations and road segments based on the diverse information contained within the images to achieve road network correction. In this task, MLLMs have to recognize not only the image content within the reference line area but also understand broader contextual information such as vehicle orientation and building layout to determine whether any junctions are missing. The reference lines for this task are sourced from OpenStreetMap [*], and the actual junction and road segment data also originate from the online map service database. The evaluation of results is divided into two parts: the accuracy of junction points and the accuracy of road segment polylines. Since the number of points and polylines returned by MLLMs may not match the ground truth, both evaluations will first match the ground truth with the MLLM output based on the nearest-neighbor principle. Points or polylines that fail to match will be considered as mapped to infinity. Point matching employs Euclidean distance, and RMSE with a distance upper bound threshold is used as the evaluation metric. For polylines, given the critical role of direction in road network correction, we employ the Fréchet Distance (Eiter et al., 1994), which evaluates directed polyline similarity, as both a distance criterion and a performance metric. This metric also incorporates a distance upper bound. Additionally, since image sizes vary, all coordinates are normalized to the range $[0, 1]$ based on the image length and width before entering the indicator calculation.

**FPV Lane Counting.** Similar to the BEV lane counting task, the lane counting task based on FPV images also requires the MLLM to determine the number of lanes from images, sharing identical evaluation metrics and ground truth data sources. The difference lies in replacing the image perspective with a first-person view captured by an in-vehicle camera. Simultaneously, the reference lines indicating the target road on the BEV image are removed, requiring MLLMs to understand the spatial relationship between the camera and the surrounding roads to make determinations.

**FPV Lane Designation Recognition.** The lane designation recognition task based on FPV images shares identical metrics and ground truth data sources with BEV lane designation recognition task. Compared to the BEV perspective, which relies entirely on road markings to identify lane direction, MLLMs in the FPV perspective can determine lane direction both through road markings and by confirming overhead signs. However, FPV images also introduce new challenges for MLLMs. Scenes captured by FPV cameras may include low-light conditions at night or situations where congested traffic obscures road markings. The real-world scenarios reflected in this data will test MLLMs' ability to synthesize information from multiple sources and produce accurate reasoning.

**FPV Road Type Classification.** The road type classification task based on FPV images requires MLLMs to determine whether the current vehicle is traveling on the main road or the service road based on the image content. Unlike other tasks, this task demands that MLLMs go beyond understanding and reasoning about a single category of urban spatial elements, such as road markings or even road networks. Instead, it requires MLLMs to grasp the semantic information implied by the surrounding environment as a whole and make inferences based on common sense. For example,

---

[*]https://www.openstreetmap.org/

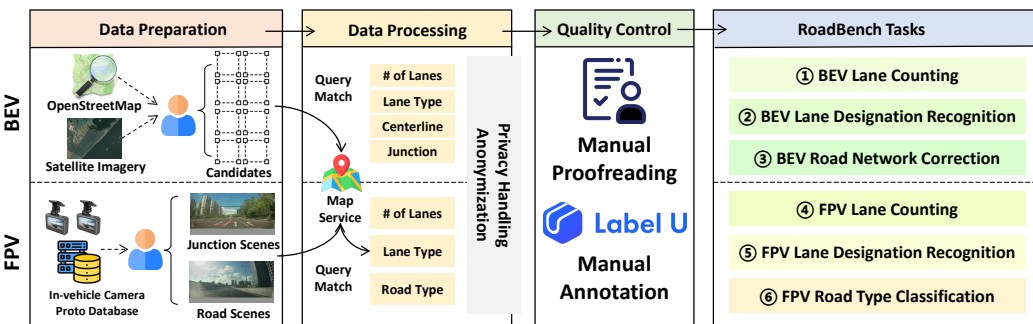

Figure 3: The RoadBench curation pipelines to construct the datasets for the six benchmark tasks.

MLLMs can determine whether a road is a service road based on pedestrians or street-side shops in an image, or identify a main road based on enclosed fences and median strips. Since the dataset construction ensures an equal distribution of test cases between the main and service roads, this constitutes a balanced binary classification task. Thus, the evaluation metric solely employs accuracy.

In summary, the aforementioned six benchmark tasks systematically evaluate MLLM's ability to understand and reason about fine-grained spatial elements in urban environments across spatial scales from local to global based on images captured from diverse perspectives and scenes.

## 3.3 BENCHMARK CURATION

The curation process for RoadBench can be divided into three stages as shown in Figure 3: data preparation, data processing, and quality control. To fully ensure data quality, most steps rely primarily on human expert annotation and rule-driven programmatic automatic matching.

**Data Preparation.** For tasks related to BEV images, the primary step in data preparation is determining appropriate spatial scopes based on satellite image resolution and junction patterns. Specifically, we first download OpenStreetMap data from areas with relatively high satellite image resolution and extract all junctions. We then manually review the corresponding satellite imagery from Google Maps [†] for each junction to eliminate invalid images that are fake, difficult to identify, or severely obscured, ultimately obtaining the final candidate bounding boxes. For FPV imagery, we extract valid instances from captured in-vehicle camera photos in junction and road scenes. For junction scene images used in lane counting and lane designation recognition, annotators are instructed to include a certain proportion of challenging scenarios such as nighttime conditions or obscured lane markings to test the capabilities of MLLMs. For road scenes, the number of images for main roads and service roads must be kept consistent to avoid class imbalance issues.

**Data Processing.** After completing data preparation, the bounding box data is fed into the database of the tier-1 online map service provider to extract labels such as actual road centerlines, junction locations, number of lanes, and lane directions within the specified area. These high-quality datasets are directly applicable for lane-related tasks. Meanwhile, when matched with OpenStreetMap data, they form inaccurate reference lines and ground truth pairs suitable for road network correction tasks. For FPV images captured at junction and road scenes, the required labels including lane count, lane direction type, and road type are matched from the map provider's database by using the image capture coordinates. All datasets have undergone anonymization and privacy handling. All IDs have been randomized into UUIDs. All location coordinates have been processed into pixel coordinates on the images. All content in FPV images that could potentially expose the privacy of the photographer or surrounding environment has been manually obscured using gray masks.

**Quality Control.** Due to the fact that certain steps involve automated program matching, a small number of annotation errors may occur. These errors can stem from factors such as poor data quality, coordinate inaccuracies, or incorrect matching. To fully ensure dataset quality, all final test cases undergo a second round of manual proofreading and are re-annotated via the LabelU annotation platform (OpenDataLab, 2025) to correct errors.

---

[†] https://www.google.com/maps

Table 2: The comprehensive evaluation results on RoadBench. The best-performing result among the evaluated models is indicated in **bold**, and the second-best result is indicated with an underline. In the table, F1, HL, and FD are abbreviations for F1-Score, Hamming Loss, and Fréchet Distance, respectively. For BEV road network correction tasks, the normalized distance upper threshold for the RMSE metric used to evaluate junction point accuracy is 20%, while the one for the Fréchet Distance metric used to evaluate segment polyline accuracy is 50%.

| Model | BEV Lane Counting | | BEV Lane Designation Recognition | | BEV Road Network Correction | | FPV Lane Counting | | FPV Lane Designation Recognition | | FPV Road Type Classification | Rank |
|---|---|---|---|---|---|---|---|---|---|---|---|---|
| | F1 ↑ | RMSE ↓ | HL ↓ | Acc. ↑ | RMSE ↓ | FD ↓ | F1 ↑ | RMSE ↓ | HL ↓ | Acc. ↑ | Acc. ↑ | |
| LLaMA-3.2-11B-Vision | 0.241 | 1.579 | 0.354 | 0.138 | 0.171 | 0.379 | 0.273 | 1.431 | 0.349 | 0.159 | 0.553 | 14 |
| LLaMA-3.2-90B-Vision | 0.295 | 1.334 | 0.207 | 0.470 | 0.149 | 0.307 | 0.305 | 1.138 | 0.161 | 0.525 | 0.640 | 6 |
| Qwen2.5-VL-7B | 0.241 | 1.409 | 0.296 | 0.103 | 0.170 | 0.398 | 0.221 | 1.416 | 0.287 | 0.167 | 0.530 | 13 |
| Qwen2.5-VL-32B | 0.280 | 1.243 | 0.226 | 0.384 | 0.154 | 0.313 | 0.335 | 1.088 | 0.232 | 0.323 | 0.563 | 11 |
| Qwen2.5-VL-72B | 0.205 | 1.342 | 0.204 | 0.477 | 0.138 | 0.259 | 0.317 | 1.201 | 0.181 | 0.481 | 0.595 | 7 |
| Gemma-3-12B | 0.222 | 1.194 | 0.250 | 0.349 | 0.159 | 0.298 | 0.220 | 1.206 | 0.237 | 0.364 | 0.575 | 12 |
| Gemma-3-27B | 0.279 | 1.331 | 0.205 | 0.415 | 0.143 | **0.232** | 0.271 | 1.206 | 0.202 | 0.404 | 0.524 | 10 |
| Gemini-2.5-Flash | 0.239 | 1.419 | 0.196 | 0.483 | 0.148 | 0.281 | 0.339 | 1.176 | 0.170 | 0.508 | 0.556 | 8 |
| Gemini-2.5-Flash-Image | 0.236 | 1.372 | 0.181 | 0.494 | 0.146 | 0.270 | 0.299 | 1.272 | 0.177 | 0.479 | 0.569 | 9 |
| Gemini-2.5-Pro | 0.322 | 1.250 | 0.164 | 0.537 | 0.144 | 0.293 | 0.509 | 0.894 | 0.143 | 0.569 | **0.764** | 3 |
| GLM-4.5V | 0.316 | 1.177 | 0.168 | 0.546 | 0.171 | 0.271 | 0.382 | 0.997 | 0.143 | 0.579 | 0.598 | 4 |
| GPT-5-Nano | 0.188 | 1.511 | 0.164 | 0.554 | **0.132** | 0.258 | 0.289 | 1.252 | 0.150 | 0.544 | 0.612 | 5 |
| GPT-5-Mini | **0.369** | 1.151 | 0.152 | 0.556 | 0.139 | 0.241 | 0.412 | 0.931 | 0.130 | 0.594 | 0.607 | **1** |
| GPT-5 | 0.309 | 1.351 | 0.156 | 0.586 | 0.142 | 0.261 | **0.526** | 0.837 | 0.129 | 0.593 | 0.705 | 2 |
| Rule-based | 0.267 | **1.109** | **0.141** | **0.605** | 0.155 | 0.381 | 0.225 | 1.440 | **0.128** | **0.602** | 0.504 | - |

## 4 EXPERIMENTS

### 4.1 EVALUATION SETTINGS

**Evaluated MLLMs.** To comprehensively evaluate the performance of MLLMs on RoadBench, we selected mainstream MLLMs released within the last one year and included both open-source and closed-source models with varying parameter counts from different providers. In the open-source models, we selected the 11B and 90B versions of LLaMA-3.2-Vision, the 7B, 32B, and 72B versions of Qwen2.5-VL (Bai et al., 2025), the 12B and 27B versions of Gemma-3 (Team et al., 2025a), and GLM-4.5V (Team et al., 2025b). In the closed-source model selection, we chose Gemini-2.5-Flash, Gemini-2.5-Flash-Image, Gemini-2.5-Pro from Google (Comanici et al., 2025), as well as GPT-5-Nano, GPT-5-Mini, and GPT-5 from OpenAI.

**Rule-based Baselines.** To aid in understanding the practical applicability of MLLM across these benchmark tasks, we include several rule-based or simple random baselines. In the two lane counting task, we provided two baselines: always choose two lanes and randomly select two lanes from $\{2, 3, 4\}$ with a uniform distribution. For the two lane designation recognition task, we designed a mapping table based on traffic common sense as a baseline. The details can be found in Appendix B.2. The FPV road type classification task uses uniformly distributed random selection as the baseline. Due to the complexity of the BEV road network correction task, we directly treat the input reference line as the output road segment polyline, using the start and end points of the reference line as the coordinates of the identified intersection. The main result shown in Table 2 will only include the optimal baseline. Complete experimental results can be found in Appendix B.

**Evaluation Metrics.** Throughout the entire experiment, the performance of MLLMs across various tasks was comprehensively evaluated using the metrics listed in Table 1. The metrics (Precision, Recall, and F1-Score) used to evaluate multi-classification tasks are weighted according to the sample size. In the BEV road network correction task, the normalized distance upper bound thresholds for RMSE and Fréchet Distance were set to $\{10\%, 20\%, 50\%\}$ to ultimately select the appropriate threshold. Due to page size limitations, the main results in Table 2 report no more than two metrics that best reflect the performance of MLLMs. The complete results can be found in Appendix B. Beyond these metrics, we also incorporate a comprehensive ranking to evaluate the relative performance of MLLMs across the entire benchmark. We first sum the rankings of all MLLMs across the two key metrics for each task, then sort this ranking sum to obtain the final rank.

**Error Handling.** To minimize the interference of detectable errors on MLLMs' fine-grained spatial understanding and reasoning evaluation, the benchmark procedure incorporates a series of error-handling mechanisms. The program detects issues such as failed API calls, empty response values, and incorrect return formats, and re-invokes MLLMs with identical inputs. This retry process is limited to a maximum of six attempts. If MLLMs still fail to produce correct results after retries, the outcome is recorded as zero or empty.

## 4.2 MAIN RESULTS

The comprehensive evaluation results on RoadBench for the selected MLLMs and baselines are listed in Table 2. By analyzing these results, we can find the following conclusions.

**RoadBench is a highly challenging benchmark for MLLMs.** Overall, neither the most powerful closed-source models nor open-source models perform sufficiently well on the tasks proposed by RoadBench for evaluating the fine-grained understanding and reasoning capabilities of MLLMs in urban scenarios. For example, in the BEV lane counting task, the best model GPT-5-Mini achieved an F1-Score of only 0.369, and the RMSE between the predicted lane counts and the ground truth exceeded 1, reaching 1.151. This indicates that MLLMs fail to effectively and robustly understand the fine-grained spatial structure of road markings in images. Furthermore, the results of the BEV road network correction task indicate that MLLMs struggle to accurately provide coordinates for corrected intersections and road segments. The best model exhibits an RMSE@20% as high as 0.132 for junction points and an FD@50% as high as 0.232 for road segment polylines, both falling within the same order of magnitude as the upper threshold set. These results suggest that current MLLMs are unable to reason based on fine-grained spatial elements and complete tasks requiring global information and also fully demonstrate that RoadBench is a highly challenging benchmark.

**MLLMs struggle to outperform simple rule-based methods that do not rely on any image inputs.** Comparing against baselines based on simple rules is a better way to understand the above results than simply interpreting the absolute values of the outcome metrics. Observing the results of the two lane designation recognition tasks reveals that none of the MLLMs can outperform the baseline designed based on traffic domain common sense. Even in the BEV viewpoint, they exhibit a relative gap of approximately 7.8% in Hamming Loss and about 3.1% in accuracy compared to the baseline. In other scenarios except the FPV road type classification task, some MLLMs also fail to outperform the baseline based on random choice. By comparing results against simple rules or random selections, we find that MLLMs still have significant room for improvement in fine-grained spatial understanding and the application of domain-specific common sense.

**MLLMs struggle to provide precise coordinate numbers.** The BEV road network correction task requires MLLMs to return coordinates for points and polylines based on their understanding and reasoning of the input. By examining the metrics shown in Table 5 at the small threshold (10%), we observe that the results are almost entirely contributed by the distance upper threshold. This indicates that the points or lines generated by MLLMs shift significantly from the ground truth, reflecting their limitations in spatial understanding and reasoning, or in the accuracy of structured numerical outputs.

**MLLMs are better at correctly understanding the fine-grained spatial elements contained within FPV images.** Comparing the results of the same task under FPV and BEV viewpoints, we observe that MLLMs demonstrate significantly superior performance on FPV images compared to BEV images, both in terms of absolute metric values and relative gaps compared to the baseline. This indicates that larger spatial elements such as road markings and signs in the FPV viewpoint can be better understood by MLLMs. Conversely, it also reflects the limitations of MLLMs in understanding more granular spatial elements in BEV images.

**The number of parameters is not a universal solution in RoadBench.** Although the number of parameters is largely positively correlated with the capabilities of MLLMs, larger models do not necessarily perform better. For example, GPT-5-Mini outperforms GPT-5 on most tasks. Qwen2.5-VL-32B also outperformed Qwen2.5-VL-72B in the FPV lane recognition task. This may be related to the fusion method or training process of the visual and textual modalities within MLLMs.

**Closed-source models hold certain technical advantages on RoadBench.** From an overall ranking perspective, in terms of fine-grained spatial understanding capabilities within urban scenarios, closed-source models represented by the GPT-5 series (Rank 1, 2, and 5) and Gemini-2.5 series (Rank 3)

have achieved certain technical advantages over open-source models. Among open-source models, only GLM-4.5V (Rank 4) ranks highly in comprehensive evaluations.

## 4.3 FURTHER ANALYSIS

**The impact of reference line prompting methods in the BEV tasks.** Since reference lines serve as the primary source for MLLMs to identify regions of interest in BEV tasks, how their positions are prompted to MLLMs may directly impact the task performance of MLLMs. We selected the closed-source model GPT-5-Mini and the open-source model GLM-4.5V, which demonstrated better performance in the BEV lane counting task and the BEV lane designation recognition task for further testing. We design different reference line prompt methods, including text-only prompts, visual-only prompts, and both text and visual prompts. Visual prompts are categorized into two approaches: using start and end point colors to indicate direction, and employing arrows on the line segments to denote direction. Both text and visual prompts with arrows are the default settings in the BEV tasks.

Based on the experimental results presented in Table 9 and Table 10, we observe significant differences in preference for prompt formats among various MLLMs, potentially attributable to variations in the data distribution used during training. GLM-4.5V shows a stronger tendency to learn from both textual prompts and image prompts with arrows, with image prompts playing a dominant role. When deprived of image prompts, GLM-4.5V's performance declines by 1.24% to 15.04%, whereas losing textual prompts only causes performance fluctuations of 0.64% to 2.29%. GPT-5-Mini shows a stronger preference for prompts that use color to distinguish directions, while variations in prompting methods produce no more than a 4% deviation in the BEV lane designation recognition task. These phenomena may indicate that GPT-5-Mini is better equipped to integrate textual and visual information to collaboratively process the entire information flow.

**The impact of scene environment conditions in the FPV tasks.** FPV images captured from in-vehicle cameras encompass varying external environmental conditions. Among these, the most direct factors affecting MLLM understanding and reasoning are adverse lighting conditions and obscured road markings. Regarding obscured road markings, only images where lane information could be determined through alternative means were retained during manual dataset proofreading. After additional annotation, 175 test cases involving adverse lighting conditions were identified, alongside 46 test cases featuring obscured road markings. To analyze the impact of different scenario environments on MLLMs, we selected GLM-4.5V, GPT-5, and Gemini-2.5-Pro which performed well in the main results as case studies.

The experimental results in the FPV lane counting task and the FPV lane designation recognition task are presented in Table 11, Table 12 and Table 13. Based on the results, we have two primary findings. First, adverse lighting conditions do indeed degrade the performance of MLLMs, but this impact is significantly alleviated in models with strong image understanding and reasoning capabilities. For example, for GPT-5, adverse lighting conditions only caused a 0.2%-1.9% drop in performance. Second, shifting the basis for task completion from obscured road markings to other elements like signage substantially improved the performance of both GPT-5 and Gemini-2.5-Pro. This phenomenon indicates that MLLMs exhibit significantly weaker capabilities in understanding and reasoning fine-grained spatial elements compared to other capabilities.

## 5 CONCLUSION

In this paper, we propose a benchmark named RoadBench with six benchmark tasks and 9,121 test cases for comprehensively evaluating MLLMs' understanding and reasoning of fine-grained spatial elements in urban scenarios based on both BEV and FPV images. Based on this benchmark, we evaluated 14 mainstream MLLMs. The results and further analysis indicate that existing MLLMs lack proper fine-grained spatial understanding and reasoning capabilities in urban scenarios. On certain tasks and metrics, they even fail to outperform baselines based on random selection or simple rules. These findings indicate the significance of RoadBench while also highlighting the need to enhance MLLM's capabilities in fine-grained spatial understanding and reasoning. Based on this fact, RoadBench is promising to become the foundational dataset and evaluation framework for advancing research and applications that enhance the fine-grained spatial understanding and reasoning capabilities of MLLM or MLLM-based agents.

## ETHICS STATEMENT

All datasets released in the benchmark were carefully hand-reviewed to ensure no leakage of collectors' or public privacy and to prevent violations of the double-blind policy. Specifically, personal view images from the crowd-sourced collection were manually inspected individually and covered with gray rectangular masks to cover text, watermarks, logos, faces, personal items, license plates, and other information that could compromise privacy or reflect the location of the data collection. Bird's-eye view images were obtained from publicly available satellite imagery published by Google. Data labels were randomized using UUIDs to ensure anonymity and to avoid associating internal data from agencies that participated in this work.

## REPRODUCIBILITY STATEMENT

To support the community in reproducing the work or using the benchmark, we have hosted the corresponding codes, example datasets, and raw results generated from MLLMs on Anonymous GitHub[‡] at `https://anonymous.4open.science/r/RoadBench-A00E`. The full dataset will be released to HuggingFace[§] after the paper has been allowed to be de-anonymized due to its excessive storage size.

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

## A  THE USE OF LARGE LANGUAGE MODELS

In this work, the use of LLMs is limited to checking for grammatical errors and providing word suggestions.

## B  ADDITIONAL EXPERIMENTAL SETUP DESCRIPTIONS AND COMPLETE RESULTS

### B.1  BEV LANE COUNTING

The complete experimental results for the BEV lane counting task are presented in Table 3. This table reports the performance of each model or baseline method based on Precision, Recall, F1-Score, and Root Mean Square Error (RMSE) metrics.

Table 3: The complete experimental results of the BEV lane counting task. The best-performing result among the evaluated models is indicated in **bold**, and the second-best result is indicated with an underline.

| Model | Precision ↑ | Recall ↑ | F1-Score ↑ | RMSE ↓ |
|---|---|---|---|---|
| LLaMA-3.2-11B-Vision | 0.2707 | 0.2586 | 0.2415 | 1.5792 |
| LLaMA-3.2-90B-Vision | 0.2904 | 0.3109 | 0.2947 | 1.3344 |
| Qwen2.5-VL-7B | 0.2718 | 0.2871 | 0.2412 | 1.4091 |
| Qwen2.5-VL-32B | 0.2882 | 0.3181 | 0.2804 | 1.2430 |
| Qwen2.5-VL-72B | 0.2994 | 0.2830 | 0.2053 | 1.3420 |
| Gemma-3-12B | 0.2529 | 0.3349 | 0.2223 | 1.1937 |
| Gemma-3-27B | 0.2817 | 0.3188 | 0.2785 | 1.3314 |
| Gemini-2.5-Flash | 0.2816 | 0.2847 | 0.2387 | 1.4192 |
| Gemini-2.5-Flash-Image | 0.2718 | 0.2892 | 0.2360 | 1.3721 |
| Gemini-2.5-Pro | 0.3286 | 0.3415 | 0.3218 | 1.2495 |
| GLM-4.5V | 0.3084 | 0.3528 | 0.3155 | 1.1765 |
| GPT-5-Nano | 0.2530 | 0.2596 | 0.1884 | 1.5107 |
| GPT-5-Mini | **0.3863** | **0.3931** | **0.3693** | 1.1515 |
| GPT-5 | 0.3365 | 0.3085 | 0.3095 | 1.3514 |
| Random Choice | 0.2498 | 0.2882 | 0.2671 | **1.1093** |
| Always 2-Lane | 0.0637 | 0.2524 | 0.1017 | 1.5323 |

### B.2  BEV LANE DESIGNATION RECOGNITION

In the lane designation recognition task, we introduce a rule-driven mapping table based on traffic common sense as a baseline:

- **1 lane:** The single lane is assigned as `left-turn`, `straight`, or `right-turn`.
- **2 lanes:** The first lane (leftmost) is designated as `left-turn` or `straight`, and the second lane (rightmost) as `straight` or `right-turn`.
- **3 or more lanes:** The first lane (leftmost) is designated as `left-turn`, the last lane (rightmost) as `right-turn`, and all intermediate lanes as `straight`.

Table 4 provides the complete results for the BEV lane designation recognition task. The models and baseline methods are evaluated using Hamming Loss and Accuracy.

### B.3  BEV ROAD NETWORK CORRECTION

The comprehensive results for the BEV road network correction task are shown in Table 5. This task evaluates model performance on junctions by RMSE and road segments by Fréchet Distance (FD) at different normalized distance upper bound thresholds.

Table 4: The complete experimental results of the BEV lane designation recognition task. The best-performing result among the evaluated models is indicated in **bold**, and the second-best result is indicated with an underline.

| Model | Hamming Loss ↓ | Acc. ↑ |
|---|---|---|
| LLaMA-3.2-11B-Vision | 0.3541 | 0.1380 |
| LLaMA-3.2-90B-Vision | 0.2075 | 0.4705 |
| Qwen2.5-VL-7B | 0.2965 | 0.1031 |
| Qwen2.5-VL-32B | 0.2258 | 0.3839 |
| Qwen2.5-VL-72B | 0.2044 | 0.4775 |
| Gemma-3-12B | 0.2496 | 0.3493 |
| Gemma-3-27B | 0.2053 | 0.4146 |
| Gemini-2.5-Flash | 0.1955 | 0.4830 |
| Gemini-2.5-Flash-Image | 0.1807 | 0.4938 |
| Gemini-2.5-Pro | 0.1640 | 0.5375 |
| GLM-4.5V | 0.1679 | 0.5461 |
| GPT-5-Nano | 0.1638 | 0.5541 |
| GPT-5-Mini | 0.1516 | 0.5557 |
| GPT-5 | 0.1557 | 0.5855 |
| Rule-based | **0.1413** | **0.6047** |

Table 5: The complete experimental results of the BEV road network correction task. The best-performing result among the evaluated models is indicated in **bold**, and the second-best result is indicated with an underline.

| Model | Junction | | | Road Segment | | |
|---|---|---|---|---|---|---|
| | RMSE@10% ↓ | RMSE@20% ↓ | RMSE@50% ↓ | FD@10% ↓ | FD@20% ↓ | FD@50% ↓ |
| LLaMA-3.2-11B-Vision | 0.0934 | 0.1707 | 0.3323 | 0.0982 | 0.1891 | 0.3792 |
| LLaMA-3.2-90B-Vision | 0.0843 | 0.1495 | 0.2720 | 0.0958 | 0.1745 | 0.3074 |
| Qwen2.5-VL-7B | 0.0923 | 0.1699 | 0.3459 | 0.0981 | 0.1881 | 0.3976 |
| Qwen2.5-VL-32B | 0.0872 | 0.1544 | 0.3185 | 0.0955 | 0.1709 | 0.3134 |
| Qwen2.5-VL-72B | 0.0813 | 0.1375 | 0.2622 | 0.0918 | 0.1570 | 0.2586 |
| Gemma-3-12B | 0.0875 | 0.1588 | 0.3179 | 0.0949 | 0.1717 | 0.2979 |
| Gemma-3-27B | 0.0852 | 0.1432 | 0.2730 | 0.0934 | 0.1537 | **0.2320** |
| Gemini-2.5-Flash | 0.0842 | 0.1482 | 0.3195 | 0.0958 | 0.1716 | 0.2808 |
| Gemini-2.5-Flash-Image | 0.0841 | 0.1462 | 0.3051 | 0.0963 | 0.1700 | 0.2700 |
| Gemini-2.5-Pro | 0.0843 | 0.1444 | 0.3065 | 0.0964 | 0.1742 | 0.2934 |
| GLM-4.5V | 0.0917 | 0.1706 | 0.3939 | **0.0899** | 0.1537 | 0.2711 |
| GPT-5-Nano | **0.0798** | **0.1319** | 0.2603 | 0.0914 | 0.1537 | 0.2575 |
| GPT-5-Mini | 0.0821 | 0.1391 | 0.2900 | 0.0907 | **0.1483** | 0.2409 |
| GPT-5 | 0.0859 | 0.1418 | 0.2813 | 0.0951 | 0.1631 | 0.2608 |
| Do Nothing | 0.0843 | 0.1554 | **0.2593** | 0.0997 | 0.1958 | 0.3807 |

## B.4 FPV LANE COUNTING

The results for the FPV lane counting task are summarized in Table 6. The table includes Precision, Recall, F1-Score, and RMSE for each model and baseline method.

Table 6: The complete experimental results of the FPV lane counting task. The best-performing result among the evaluated models is indicated in **bold**, and the second-best result is indicated with an underline.

| Model | Precision ↑ | Recall ↑ | F1-Score ↑ | RMSE ↓ |
|---|---|---|---|---|
| LLaMA-3.2-11B-Vision | 0.2717 | 0.2944 | 0.0000 | 1.4314 |
| LLaMA-3.2-90B-Vision | 0.4003 | 0.3636 | 0.0000 | 1.1383 |
| Qwen2.5-VL-7B | 0.3350 | 0.2822 | 0.2207 | 1.4161 |
| Qwen2.5-VL-32B | 0.3978 | 0.3935 | 0.3354 | 1.0884 |
| Qwen2.5-VL-72B | 0.4720 | 0.3826 | 0.3165 | 1.2010 |
| Gemma-3-12B | 0.3694 | 0.2931 | 0.2199 | 1.2055 |
| Gemma-3-27B | 0.3345 | 0.3134 | 0.2711 | 1.2060 |
| Gemini-2.5-Flash | 0.4318 | 0.3758 | 0.3392 | 1.1759 |
| Gemini-2.5-Flash-Image | 0.4220 | 0.3569 | 0.2989 | 1.0000 |
| Gemini-2.5-Pro | 0.5399 | 0.5183 | 0.5093 | 0.8940 |
| GLM-4.5V | 0.4746 | 0.4179 | 0.3817 | 0.9973 |
| GPT-5-Nano | 0.4236 | 0.3677 | 0.2890 | 1.2519 |
| GPT-5-Mini | 0.4909 | 0.4532 | 0.4117 | 0.9311 |
| GPT-5 | **0.5652** | **0.5455** | **0.5261** | **0.8367** |
| Random Choice | 0.2065 | 0.2497 | 0.2245 | 1.4404 |
| Always 2-Lane | 0.0644 | 0.2537 | 0.1027 | 1.6111 |

## B.5 FPV LANE DESIGNATION RECOGNITION

Table 7 displays the complete experimental outcomes for the FPV lane designation recognition task, measured by Hamming Loss and Accuracy.

Table 7: The complete experimental results of the FPV lane designation recognition task. The best-performing result among the evaluated models is indicated in **bold**, and the second-best result is indicated with an underline.

| Model | Hamming Loss ↓ | Acc. ↑ |
|---|---|---|
| LLaMA-3.2-11B-Vision | 0.3492 | 0.1586 |
| LLaMA-3.2-90B-Vision | 0.1609 | 0.5246 |
| Qwen2.5-VL-7B | 0.2869 | 0.1669 |
| Qwen2.5-VL-32B | 0.2316 | 0.3233 |
| Qwen2.5-VL-72B | 0.1809 | 0.4807 |
| Gemma-3-12B | 0.2369 | 0.3642 |
| Gemma-3-27B | 0.2019 | 0.4037 |
| Gemini-2.5-Flash | 0.1703 | 0.5085 |
| Gemini-2.5-Flash-Image | 0.1773 | 0.4794 |
| Gemini-2.5-Pro | 0.1429 | 0.5693 |
| GLM-4.5V | 0.1430 | 0.5793 |
| GPT-5-Nano | 0.1500 | 0.5441 |
| GPT-5-Mini | 0.1301 | 0.5941 |
| GPT-5 | 0.1287 | 0.5932 |
| Rule-based | **0.1282** | **0.6019** |

### B.6 FPV ROAD TYPE CLASSIFICATION

The performance of all models on the FPV road type classification task is detailed in Table 8, with results reported in Accuracy.

Table 8: The complete experimental results of the FPV road type classification task. The best-performing result among the evaluated models is indicated in **bold**, and the second-best result is indicated with an underline.

| Model | Acc. ↑ |
|---|---|
| LLaMA-3.2-11B-Vision | 0.5530 |
| LLaMA-3.2-90B-Vision | 0.6398 |
| Qwen2.5-VL-7B | 0.5298 |
| Qwen2.5-VL-32B | 0.5631 |
| Qwen2.5-VL-72B | 0.5954 |
| Gemma-3-12B | 0.5752 |
| Gemma-3-27B | 0.5237 |
| Gemini-2.5-Flash | 0.5560 |
| Gemini-2.5-Flash-Image | 0.5691 |
| Gemini-2.5-Pro | **0.7639** |
| GLM-4.5V | 0.5984 |
| GPT-5-Nano | 0.6115 |
| GPT-5-Mini | 0.6065 |
| GPT-5 | 0.7053 |
| Random Choice | 0.5045 |

## C RESULTS OF FURTHER ANALYSIS

### C.1 THE IMPACT OF REFERENCE LINE PROMPTING METHODS IN THE BEV TASKS

The experimental results of the impact of reference line prompting methods in the BEV tasks are shown in Table 9 and Table 10. For all results, the relative change ratios of metrics for other prompting methods were calculated relative to the benchmark default settings (both prompts with arrows) to visualize the differences.

Table 9: GLM-4.5V performance comparison with different prompt strategies. The best-performing result among the evaluated models is indicated in **bold**, and the second-best result is indicated with an underline.

| Method | BEV Lane Counting | | BEV Lane Designation Recognition | |
|---|---|---|---|---|
| | F1-Score ↑ | RMSE ↓ | Hamming Loss ↓ | Acc. ↑ |
| Text-only | 0.3116 (-1.24%) | 1.3535 (+15.04%) | 0.1847 (+10.01%) | 0.4924 (-9.83%) |
| Visual-only (Color) | **0.3184 (+0.92%)** | 1.1904 (+1.18%) | 0.1701 (+1.31%) | 0.5389 (-1.32%) |
| Visual-only (Arrow) | 0.3088 (-2.12%) | **1.1496 (-2.29%)** | 0.1694 (+0.89%) | 0.5426 (-0.64%) |
| Both (Color) | 0.3131 (-0.76%) | 1.2820 (+8.97%) | 0.1697 (+1.07%) | 0.5439 (-0.40%) |
| Both (Arrow) | 0.3155 (-) | 1.1765 (-) | **0.1679 (-)** | **0.5461 (-)** |

### C.2 THE IMPACT OF SCENE ENVIRONMENT CONDITIONS IN THE FPV TASKS

The experimental results of the impact of scene environment conditions in the FPV tasks are shown in Table 11, Table 12, and Table 13. For all results, the relative change ratios of metrics for other prompting methods were calculated relative to those of the full datasets to visualize the differences.

Table 10: GPT-5-Mini performance comparison with different prompt strategies. The best-performing result among the evaluated models is indicated in **bold**, and the second-best result is indicated with an underline.

| Method | BEV Lane Counting | | BEV Lane Designation Recognition | |
|---|---|---|---|---|
| | F1-Score ↑ | RMSE ↓ | Hamming Loss ↓ | Acc. ↑ |
| Text-only | 0.3379 (-8.50%) | 1.2419 (+7.85%) | 0.1491 (-1.65%) | 0.5613 (+1.01%) |
| Visual-only (Color) | 0.3606 (-2.36%) | 1.1867 (+3.06%) | 0.1467 (-3.23%) | 0.5657 (+1.80%) |
| Visual-only (Arrow) | 0.3500 (-5.23%) | 1.1671 (+1.35%) | 0.1478 (-2.51%) | 0.5679 (+2.20%) |
| Both (Color) | 0.3569 (-3.36%) | **1.1428 (-0.76%)** | **0.1465 (-3.36%)** | **0.5682 (+2.25%)** |
| Both (Arrow) | **0.3693 (-)** | 1.1515 (-) | 0.1516 (-) | 0.5557 (-) |

Table 11: GLM-4.5V performance analysis across different environmental conditions.

| Condition | FPV Lane Counting | | FPV Lane Designation Recognition | |
|---|---|---|---|---|
| | F1-Score ↑ | RMSE ↓ | Hamming Loss ↓ | Accuracy ↑ |
| Adverse Lighting Conditions | 0.3283 (-14.0%) | 1.0170 (+2.0%) | 0.1509 (+5.5%) | 0.5434 (-6.2%) |
| Obscured Road Markings | 0.3838 (+0.6%) | 1.0632 (+6.6%) | 0.1271 (-11.1%) | 0.5543 (-4.3%) |
| **All** | 0.3817 (-) | 0.9973 (-) | 0.1430 (-) | 0.5793 (-) |

Table 12: GPT-5 performance analysis across different environmental conditions.

| Condition | FPV Lane Counting | | FPV Lane Designation Recognition | |
|---|---|---|---|---|
| | F1-Score ↑ | RMSE ↓ | Hamming Loss ↓ | Accuracy ↑ |
| Adverse Lighting Conditions | 0.5272 (+0.2%) | 0.8485 (+1.4%) | 0.1311 (+1.9%) | 0.5887 (-0.8%) |
| Obscured Road Markings | 0.4674 (-11.2%) | 0.9555 (+14.2%) | 0.1371 (+6.5%) | 0.5543 (-6.6%) |
| **All** | 0.5261 (-) | 0.8367 (-) | 0.1287 (-) | 0.5932 (-) |

Table 13: Gemini-2.5-Pro performance analysis across different environmental conditions.

| Condition | FPV Lane Counting | | FPV Lane Designation Recognition | |
|---|---|---|---|---|
| | F1-Score ↑ | RMSE ↓ | Hamming Loss ↓ | Accuracy ↑ |
| Adverse Lighting Conditions | 0.4932 (-3.2%) | 0.9442 (+5.6%) | 0.1462 (+2.3%) | 0.5491 (-3.5%) |
| Obscured Road Markings | 0.6369 (+25.1%) | 0.7518 (-15.9%) | 0.1357 (-5.0%) | 0.5257 (-7.7%) |
| **All** | 0.5093 (-) | 0.8940 (-) | 0.1429 (-) | 0.5693 (-) |

# D  PROMPTS OF BENCHMARK TASKS

Examples of prompts for each task in RoadBench and typical MLLM responses are listed below. It is worth noting that in the prompt design for RoadBench, the requirement for MLLM to return YAML format with a line of comments balances the need for explicit reasoning with the demand for structured output.

## D.1  BEV LANE COUNTING

An example image (Figure 4) and prompt of the BEV lane counting task are as follows:

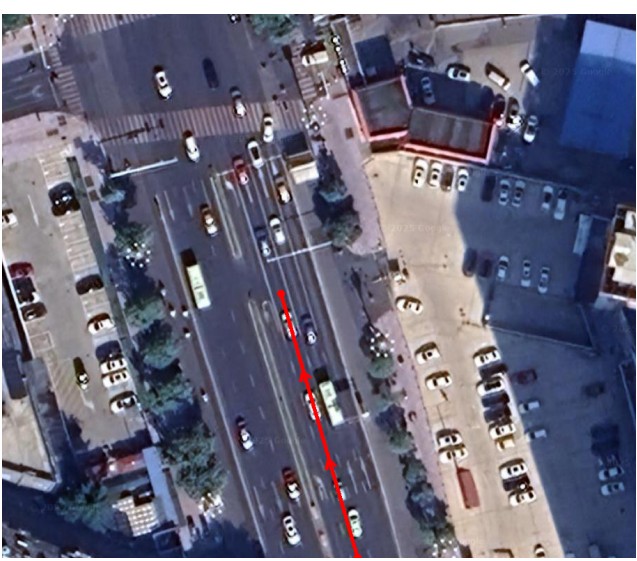

Figure 4: An example image input to MLLM in the BEV lane counting task.

**Task: Lane Counting for Road Network Analysis**
**Image Description:** This is a satellite image of a road network in China, where vehicles drive on the right side of the road. A specific road segment has been marked as a red polyline with arrowheads.
**Data:**
```yaml
# The image size
image:
  width: 910
  height: 779
# The pixel coordinates of the reference centerline
coordinates:
- x: 510.0
  y: 778.0
- x: 400.0
  y: 400.0
```
**Question:** Analyze the marked road segment and determine: How many lanes are available for vehicles traveling in the direction of the arrow?
**YAML Output Requirements:**
  • Add ONLY one-line YAML comment explaining your visual analysis and reasoning
  • Extract num_lanes (integer)
**Output Format:**
```yaml

```
# Explanation: [Describe what you see that led to your determination
    ]
num_lanes: <integer>
```
Your YAML output:
```

Typical MLLM responses for the image and prompt are as follows:

**Response from GPT-5:**
```yaml
# Explanation: On the right half of the divided roadway (arrow
    direction), there are three clearly marked lanes separated by
    dashed white lines between the curb and the central median.
num_lanes: 3
```
**Response from Qwen2.5-7B-instruct:**
```yaml
Explanation: Observing from top-down view, there appear three
    distinct parallel lines indicating separate traffic lanes.
num_lanes: 3
```

## D.2   BEV LANE DESIGNATION RECOGNITION

An example image (Figure 5) and prompt of the BEV lane designation recognition task are as follows:

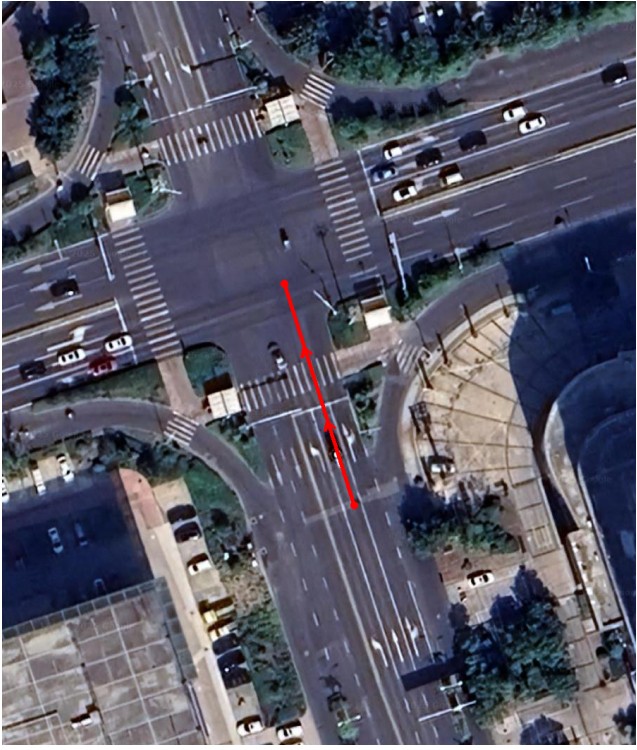

Figure 5: An example image input to MLLM in the BEV lane designation recognition task.

**Task: Lane Designations Recognition for Road Network Analysis**
**Image Description:** This is a satellite image of a road network in China, where vehicles drive on the right side of the road. A specific road segment has been marked as a red polyline with arrowheads.
**Data:**
```yaml
# The number of lanes
num_lanes: 3
# The image size
image:
  width: 899
  height: 1054
# The pixel coordinates of the reference centerline
coordinates:
- x: 499.0
  y: 714.0
- x: 400.0
  y: 400.0
```
**Question:** Analyze the marked road segment and determine: What are the lane designations in the direction of the arrow?
**Lane Direction Types:** The available lane direction types include:
(1) U-turn
(2) left-turn
(3) straight
(4) right-turn
**YAML Output Requirements:**
- Add ONLY one-line YAML comment explaining your visual analysis and reasoning
- Return `lane_designations` as a list where each item represents the lane direction types for each lane from left to right
    - Make sure the number of lane designations is equal to the number of lanes (which is 3)
- Each lane can have multiple direction types, so each item in the list should also be a list
- Use the English terms: `"U-turn"`, `"left-turn"`, `"straight"`, `"right-turn"`
**Output Format:**
```yaml
# Explanation: [Describe what you see that led to your determination
    ]
lane_designations: [["direction1", "direction2"], ["direction3"],
    ...]
```
Your YAML output:

Typical MLLM responses for the image and prompt are as follows:

**Response from GPT-5-mini:**
```
# Explanation: Pavement arrow markings show a dedicated left-turn
    lane on the left, a straight-only center line, and a rightmost
    lane marked for both straight and right turns
lane_designations: [["left-turn"], ["straight"], ["straight", "right
    -turn"]]
```
**Response from Google Gemini-2.5-Flash:**
```yaml
# Explanation: From the bottom of the marked segment, the leftmost
    lane has a straight arrow and a right-turn arrow. The middle
    lane has a straight arrow. The rightmost lane has a straight
    arrow.
```

```
lane_designations: [["straight", "right-turn"], ["straight"], ["
    straight"]]
```

### D.3 BEV ROAD NETWORK CORRECTION

An example image (Figure 6) and prompt of the BEV road network correction task are as follows:

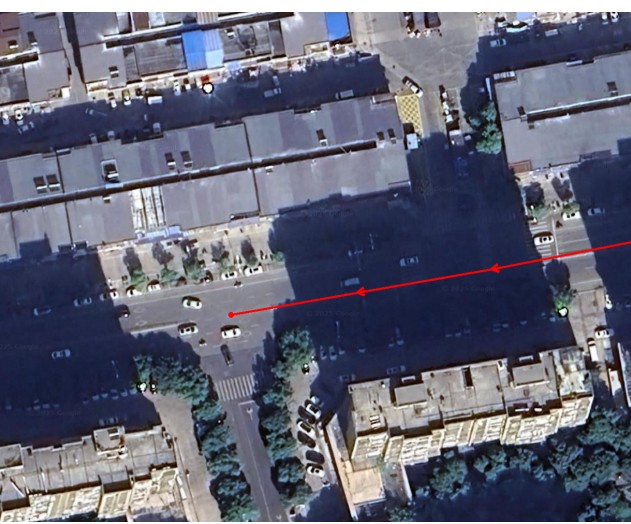

Figure 6: An example image input to MLLM in the BEV road network correction task.

**Task: Road Network Modification and Junction Identification**
**Image Description:** This is a satellite image of a road network in China, where vehicles drive on the right side of the road. A reference line has been marked as a red directed polyline to indicate the path and direction of travel along a road.
**Data:**
```yaml
# The image size
image:
  width: 1225
  height: 984
# The pixel coordinates of the reference centerline
coordinates:
- x: 1225.0
  y: 443.0
- x: 958.0
  y: 495.0
- x: 448.0
  y: 584.0
```
**Task Description:** The given reference line may have missed important junctions such as intersections, highway on/off ramps, U-turn locations, dedicated right-turn lanes, etc., and it may not always be accurately aligned with the true centerline of the road. Your task is to analyze the image and identify:
1. **Junctions**: Important road intersections, merging points, or decision points that should be marked. Junctions are represented by WKT POINT geometries.
2. **Line Segments (with Directionality)**: Road segments between junctions that represent the centerlines of the roads, with attention to the direction of travel (from start to end, following the

reference line's direction). Line segments are represented by WKT LINESTRING geometries, and their order and orientation should reflect the actual direction of traffic flow along the road.

**Analysis Requirements:**
- Carefully examine the directed reference line and the actual road network in the image
- Identify any missing junctions where the directed reference line should be split (intersections, ramps, etc.)
- Return corrected directed line segments that represent the centerlines between identified junctions
- Ensure junctions are placed at the center points of intersections/decision points
- Ensure line segments follow the actual road centerlines
- Do not return junctions or line segments that are not related to the directed reference line

**YAML Output Requirements:**
- Add ONLY one-line YAML comment explaining your analysis and corrections
- Return `junctions` as a list of WKT POINT geometries representing junction center points
- Return `lines` as a list of directed WKT LINESTRING geometries representing road centerlines between junctions
- Use pixel coordinates in the format: `"POINT (x y)"` and `"LINESTRING (x1 y1, x2 y2, ...)"`

**Output Format:**
```yaml
# Explanation: [Describe the junctions and line segments you
    identified and any corrections made to the directed reference
    line]
junctions: ["POINT (x1 y1)", "POINT (x2 y2)", ...]
lines: ["LINESTRING (x1 y1, x2 y2)", "LINESTRING (x2 y2, x3 y3)",
    ...]
```
Your YAML output:

Typical MLLM responses for the image and prompt are as follows:

**Response from Qwen2.5-VL-72B-instruct:**
```yaml
# Explanation: The reference line was extended to include a missed
    junction at the intersection and corrected to follow the road
    centerline accurately.
junctions: ["POINT (1225 443)", "POINT (958 495)", "POINT (700 550)
    ", "POINT (448 584)"]
lines: ["LINESTRING (1225 443, 958 495)", "LINESTRING (958 495, 700
    550)", "LINESTRING (700 550, 448 584)"]
```

**Response from Llama-3.2-11B-vision-instruct:**
```
# Explanation: Corrected and completed the directed reference line,
    identifying all junctions and centerlines between them, ensuring
     accurate representation of the road network in China.

junctions: ["POINT (958.0 495.0)", "POINT (448.0 584.0)", "POINT
    (1225.0 443.0)"]
lines: ["LINESTRING (1225.0 443.0, 958.0 495.0)", "LINESTRING (958.0
     495.0, 448.0 584.0)", "LINESTRING (448.0 584.0, 1225.0 443.0)"]
```

### D.4 FPV LANE COUNTING

An example image (Figure 7) and prompt of the FPV lane counting task are as follows:

Figure 7: An example image input to MLLM in the FPV lane counting task.

---

**Task: Lane Counting for Road Network Analysis**
**Image Description:** This is a personal viewpoint image of a road in China, where vehicles drive on the right side of the road.
**Question:** Analyze the image and determine: How many lanes are available for vehicles traveling in the direction of the road?
**YAML Output Requirements:**
- Add ONLY one-line YAML comment explaining your visual analysis and reasoning
- Extract num_lanes (integer)

**Output Format:**
```yaml
# Explanation: [Describe what you see that led to your determination
    ]
num_lanes: <integer>
```
Your YAML output:

---

Typical MLLM responses for the image and prompt are as follows:

---

**Response from Llama-3.2-90B-vision-instruct:**
```yaml
# The image shows a road with a clear division of lanes through
    white markings, indicating three distinct lanes for vehicles to
    travel in the direction shown.
num_lanes: 3
```
**Response from Qwen2.5-VL-7B-instruct:**
```yaml
# Explanation: The image shows two solid white lines marking
    separate travel paths.
num_lanes: 2
```

---

## D.5 FPV LANE DESIGNATION RECOGNITION

An example image (Figure 8) and prompt of the FPV lane designation recognition task are as follows:

---

**Task: Lane Designations Recognition for Road Network Analysis**
**Image Description:** This is a personal viewpoint image of a road in China, where vehicles drive on the right side of the road.

---

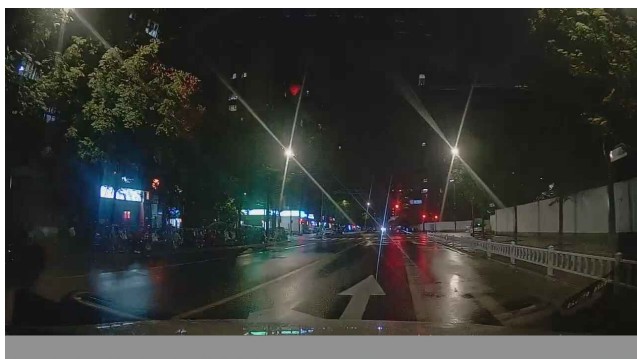

Figure 8: An example image input to MLLM in the FPV lane designation recognition task.

**Data:**
```yaml
# The number of lanes
num_lanes: 2
```
**Question:** Analyze the image and determine: What are the lane designations in the direction of the road?
**Lane Direction Types:** The available lane direction types include:
(1) U-turn
(2) left-turn
(3) straight
(4) right-turn
(5) variable
**YAML Output Requirements:**
- Add ONLY one-line YAML comment explaining your visual analysis and reasoning
- Return `lane_designations` as a list where each item represents the lane direction types for each lane from left to right
  - Make sure the number of lane designations is equal to the number of lanes (which is 2)
- Each lane can have multiple direction types, so each item in the list should also be a list
- Use the English terms: `"U-turn"`, `"left-turn"`, `"straight"`, `"right-turn"`, `"variable"`
**Output Format:**
```yaml
# Explanation: [Describe what you see that led to your determination
    ]
lane_designations: [["direction1", "direction2"], ["direction3"],
    ...]
```
Your YAML output:

Typical MLLM responses for the image and prompt are as follows:

**Response from GLM-4.5V:**
```yaml
# Explanation: The image shows a road with two lanes; the visible
    lane marking indicates a straight direction, and there are no
    other directional markings (like left-turn or right-turn arrows)
     for either lane, so both lanes are designated for straight
    travel.
```

```
lane_designations: [["straight"], ["straight"]]
```
**Response from GPT-5-nano:**
```
# Explanation: The two lanes ahead have forward (straight) arrow
    markings with no visible left/right/U-turn arrows, indicating
    straight travel for both lanes
lane_designations: [["straight"], ["straight"]]
```

### D.6 FPV ROAD TYPE CLASSIFICATION

An example image (Figure 9) and prompt of the FPV road type classification task are as follows:

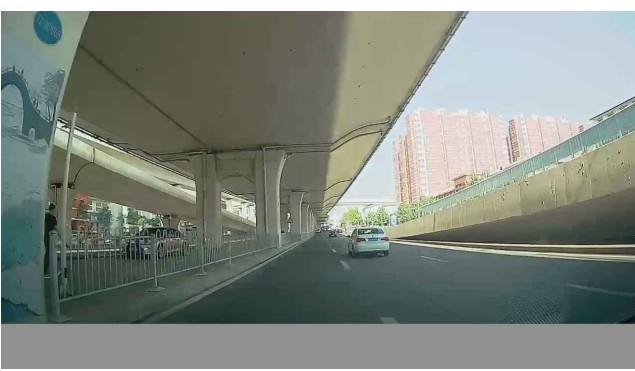

Figure 9: An example image input to MLLM in the FPV road type classification task.

**Task: Road Type Classification**
**Image Description:** This is a personal viewpoint image of a road in China, where vehicles drive on the right side of the road.
**Question:** Analyze the image and determine: Is the current vehicle traveling on a main road or a service road?
**Road Type Definitions:**
- **Main road**: Primary traffic arteries, highways, major streets with higher traffic capacity and speed limits
- **Service road**: Secondary roads, access roads, frontage roads, or auxiliary roads that typically serve adjacent properties or provide access to main roads
**YAML Output Requirements:**
- Add ONLY one-line YAML comment explaining your visual analysis and reasoning
- Extract road_type as either "main" or "service"
**Output Format:**
```yaml
# Explanation: [Describe what you see that led to your determination
    ]
road_type: <"main" or "service">
```
Your YAML output:

Typical MLLM responses for the image and prompt are as follows:

**Response from Qwen2.5-VL-32B-instruct:**
```yaml
# Explanation: The presence of multiple lanes for vehicular movement
    , clear lane markings, and an elevated structure (likely part of
```

```
     a highway overpass) suggest this is a primary roadway designed
   for high-capacity traffic flow.
road_type: main
```
```

**Response from Gemma-3-12B-it:**
```yaml
# The presence of multiple lanes, a highway overpass, distinct
   separation from adjacent buildings, and steady vehicle flow
   suggest this is a main road.
road_type: main
```
```

