# OpenReview forum: "RoadBench: Benchmarking MLLMs on Fine-Grained Spatial Understanding and Reasoning in Urban Road Scenarios"
_ICLR.cc/2026/Conference — ICLR 2026 Conference Withdrawn Submission_

### Official Review · Reviewer_8ttf · 2025-10-31

**Soundness:** 3
**Presentation:** 3
**Contribution:** 2
**Rating:** 4
**Confidence:** 3

**Summary:**

This paper proposes RoadBench, a benchmark for evaluating MLLMs’ fine-grained spatial understanding and reasoning capabilities with respect to road markings and urban traffic systems. Specifically, it comprises six tasks and includes 9,121 strictly manually verified test cases, spanning from local spatial understanding to global reasoning. Extensive comparisons across 14 MLLMs and rule-based or random-selection baselines show that RoadBench is a challenging benchmark, and these findings provide guidance for advancing MLLMs' spatial understanding.

**Strengths:**

- The writing is quite clear and it is easy to follow.
- The motivation for evaluating the real-world spatial understanding and reasoning capabilities in MLLMs is reasonable.
- The main results and further analysis are informative and interesting.
- The code, example datasets, and raw evaluation results are provided.

**Weaknesses:**

Although the paper is generally well-presented, I do have a few concerns and minor suggestions:

- My main concern is about the design of the six tasks in RoadBench:
    - Some tasks, like Lane Counting and Lane Designation Recognition, feel a bit too simple. In fact, the rule-based baselines outperform most MLLMs, which suggests these tasks might not require deep reasoning at all. With just a bit of task-specific fine-tuning, MLLMs could likely do much better. Given this, it’s hard to call RoadBench a highly challenging benchmark overall.
    - Moreover, nearly all tasks focus only on lane markings. But from a first-person view, urban scenes contain many other rich cues, like traffic signs, signals, and road boundaries, that are essential for understanding traffic systems. Relying solely on lane markings makes the benchmark feel narrow and repetitive, and it’s unclear whether it truly captures fine-grained spatial understanding in complex urban scenarios as claimed.
- Line 197-199: The paper states that test cases cover diverse scenarios (different road/intersection patterns, lighting conditions, seasons, and image resolutions). Please provide a table or figure showing the actual distribution or statistics to demonstrate this coverage.
- Lines 445-446 and 465-466: Tables 9-13 are mentioned here but don’t appear in the main text. Please clarify that they’re in the Appendix (e.g., “see Table 9 in Appendix”) to avoid confusion.

**Questions:**

Please see the Weaknesses

---

### Official Review · Reviewer_ABfH · 2025-10-31

**Soundness:** 2
**Presentation:** 2
**Contribution:** 2
**Rating:** 4
**Confidence:** 4

**Summary:**

This paper introduces RoadBench, a new benchmark with 9,121 test cases designed to evaluate the fine-grained spatial understanding of MLLMs in urban environments. The benchmark's core premise is to test model capabilities on fine-grained elements, specifically road markings (like lane lines, directional arrows) in urban settings. The data is sourced from real-world sources including satellite imagery for Bird's-Eye View (BEV) and crowd-sourced in-vehicle camera databases for First-Person View (FPV). The benchmark is structured into six tasks, covering both BEV and FPV perspectives: BEV Lane Counting, BEV Lane Designation Recognition, BEV Road Network Correction, FPV Lane Counting, FPV Lane Designation Recognition, and FPV Road Type Classification.

The paper evaluates 14 mainstream MLLMs, including the GPT-5, Gemini, and Qwen series. The primary finding is that these SOTA models perform very poorly, and in many cases, fail to outperform a rule-based baselines that do not use the image priors.

**Strengths:**

1. The benchmark is well-structured, building in complexity from local perception (lane counting) to global reasoning (road network correction). The use of both different viewpoints for different tasks is interesting.
2. With 9,121 test cases, the dataset is of a substantial size. The data is manually verified against high-quality ground truth from a commercial map service provider.

**Weaknesses:**

1. The paper states data is only from Chinese cities. This is a significant limitation, as road marking standards (colors, line patterns, arrow shapes, text) vary dramatically between countries. The benchmark may only be evaluating performance on one specific set of traffic standards, and the findings might not generalize globally.
2. While 14 models are tested, they are variants from only six model families (LLaMA, Qwen, Gemma, Gemini, GLM, GPT). The conclusions about MLLMs' failures would be stronger if tested against an even more diverse set of architectures.
3. The rule-based baseline is not very meaningful. The baseline predicts 2 or samples from {2,4,6} for the lane counting task. This does not seem methodological, the authors should instead establish a human performance baseline by recruiting humans to test their performance on this task.
4. The lane correction task is not very well-structured. The task requires the MLLM to output exact coordinates in a strict POINT (x y) format. A model might correctly identify the missing junction (good spatial reasoning) but then fail to accurately identify the coordinate string. Additionally, the task asks for the "center point" of a junction which can be ambiguous.

**Questions:**

1. How many "human experts" were involved, and what specific instructions or annotation guidelines were they given?
2. How were the markings/arrows added to the images?
3. What was the resolution of input images?

---

### Official Review · Reviewer_ENUw · 2025-11-01

**Soundness:** 3
**Presentation:** 3
**Contribution:** 3
**Rating:** 6
**Confidence:** 4

**Summary:**

This manuscript introduces RoadBench, a systematic benchmark designed to evaluate multimodal large language models (MLLMs) on fine-grained spatial understanding and reasoning tasks within urban road environments.

The authors argue that existing benchmarks primarily target global scene understanding or isolated object recognition, leaving fine-grained spatial reasoning, especially over structured urban elements like road markings, largely unexplored.

RoadBench provides 9,121 manually verified test cases across six tasks (lane counting, lane designation recognition, road network correction, and road type classification) using both Bird’s-Eye View (BEV) and First-Person View (FPV) imagery. It evaluates 14 open- and closed-source MLLMs (e.g., GPT-5, Gemini-2.5, Qwen2.5-VL, GLM-4.5V), revealing that current models often fail to outperform simple rule-based baselines on fine-grained spatial reasoning.

The benchmark pipeline, data sources (OpenStreetMap, Google Maps, in-vehicle imagery), and evaluation metrics (Precision, Recall, F1, RMSE, Hamming Loss, Fréchet Distance) are clearly described, and the benchmark will be publicly released for reproducibility.

**Strengths:**

(+) The authors convincingly articulate the gap between high-level spatial reasoning (e.g., DriveBench, CityBench) and fine-grained spatial understanding (e.g., interpreting lane markings, small-scale topologies). The focus on road markings as symbolic spatial primitives is novel and intuitively important.

(+) The six tasks in RoadBench collectively cover both local (lane-level) and global (network-level) reasoning, forming a coherent evaluation hierarchy rarely seen in current MLLM benchmarks. Manual annotation, privacy masking, and multi-source verification lend credibility to the benchmark’s rigor. The inclusion of both BEV and FPV perspectives broadens applicability.

(+) The manuscript evaluates a diverse set of recent MLLMs under unified settings, providing a useful snapshot of current progress and limitations in fine-grained spatial reasoning.

**Weaknesses:**

(-) The contribution lies primarily in benchmark construction and empirical evaluation rather than algorithmic innovation. While this is acceptable for a benchmark paper, a deeper methodological discussion (e.g., design rationale for task hierarchy or metric selection) would strengthen the manuscript’s impact.

(-) Narrow domain scope. All data come from Chinese cities and specific imagery sources. Although diverse in lighting and weather, the benchmark may lack geographical and cultural diversity, potentially limiting generalizability.

(-) While the manuscript cites CityBench, DriveBench, and UrBench, a more quantitative or conceptual comparison (e.g., task type, spatial granularity, reasoning depth) is needed to clearly position RoadBench’s uniqueness.

(-) Missing ablation or diagnostic analysis on benchmark difficulty. Although performance results show low scores, it is unclear which factors (e.g., visual ambiguity, text-format sensitivity, prompt design) contribute most to failures. Some case-level breakdowns would add insight.

**Questions:**

In addition to the concerns raised in the above section, please clarify the following questions:

- Clarify whether BEV and FPV tasks share identical data splits or were independently curated.

- Add a visual comparison figure showing sample tasks from CityBench, DriveBench, and RoadBench for clearer contrast.

- Report inter-annotator consistency or error rate in manual verification for completeness.

- Consider including a short discussion on how RoadBench could support training or instruction-tuning, not only evaluation.

- Some sections (e.g., Table 2 description) could be more concise; the manuscript is slightly verbose.

---

### Official Review · Reviewer_UASN · 2025-11-01

**Soundness:** 2
**Presentation:** 2
**Contribution:** 2
**Rating:** 4
**Confidence:** 3

**Summary:**

This paper proposes s a new benchmark targeting multimodal large language models (MLLMs) and their ability to understand fine-grained spatial details in urban road scenes It specifically focuses on road markings – the painted lanes, arrows, and symbols on road surfaces that organize traffic. RoadBench comprises 6 tasks (with 9,121 total test cases) that range from local perception to global reasoning. These tasks use two image perspectives: bird’s-eye view (BEV) images (satellite maps) and first-person view (FPV) images.

**Strengths:**

MLLMs has been widely utilized in autonomous driving area, yet the spatial understanding ability of different MLLM are not yet very well investiagated. Having more dataset, benchmark to evaluate the model ability is crucial or undertand the MLLM's reliablity.

**Weaknesses:**

1. We've observed a bunch of works in this area, like [1,2,3] have similar intentions. Have many benchmarks nnd dataset is not a big issue for an area. However, the main paper hasn't clearly stated the difference between this new benchmark and previous benchmarks.

2. The different task setting for evaluation is good to test various of fileds, however, when it comes to the final score of the benchmark ranking, are the final score the mean value of sub-tasks? I do think the paper lacks a detailed description of how the evaluation metrics are designed. According to my personal experience when designing the benchmark, the fairness could be a core issue. Some LLMs are more get fit into specific formation of questions although the evaluation target is the same. How authors design to avoid fairness issue is not clear in this version.

3. Normally, benchmark paper could do some baseline post-training experiment to help people understand how hard the benchmark is, this is also a lacking part.


[1] SPACE: Evaluating Spatial Cognition in Frontier Models (ICLR 2025)

[2] SURDS: Benchmarking Spatial Understanding and Reasoning in Driving Scenarios with Vision Language Models

[3] CityEQA: Embodied Question Answering in Cities

**Questions:**

-

---

### Note · Authors · 2025-11-12

I have read and agree with the venue's withdrawal policy on behalf of myself and my co-authors.